# Breast Cancer Molecular Subtypes Differentially Express Gluconeogenic Rate-Limiting Enzymes—Obesity as a Crucial Player

**DOI:** 10.3390/cancers15204936

**Published:** 2023-10-11

**Authors:** Carla Luís, Fernando Schmitt, Rute Fernandes, Nuno Coimbra, Joana Rigor, Paula Dias, Dina Leitão, Rúben Fernandes, Raquel Soares

**Affiliations:** 1Biochemistry Unit, Department of Biomedicine, Faculty of Medicine, University of Porto (FMUP), Al. Prof. Hernâni Monteiro, 4200-319 Porto, Portugal; carlaluis@med.up.pt (C.L.); joanarigor@gmail.com (J.R.); 2i3S—Instituto de Inovação e Investigação em Saúde, University of Porto, 4200-135 Porto, Portugal; ruben.fernandes@ufp.edu.pt; 3Institute of Molecular Pathology and Immunology of the University of Porto (IPATIMUP), 4200-135 Porto, Portugal; fschmitt@ipatimup.pt; 4Pathology and Oncology Unit, Pathological Anatomy Laboratory, Faculty of Medicine, University of Porto (FMUP), Al. Prof. Hernâni Monteiro, 4200-319 Porto, Portugal; dinaraquel@med.up.pt; 5CINTESIS@RISE, Health Research Network, 4200-319 Porto, Portugal; 6Portuguese Oncology Institute of Porto, 4200-072 Porto, Portugal; rute.j.fernandes@gmail.com (R.F.); nuno.coimbra@ipoporto.min-saude.pt (N.C.); 7Cancer Biology & Epigenetics Group, Research Center of IPO Porto (CI-IPOP)/RISE@CI-IPOP (Health Research Network), Portuguese Oncology Institute of Porto (IPO-Porto)/Porto Comprehensive Cancer Center Raquel Seruca (Porto.CCC), 4200-072 Porto, Portugal; paulacmdias@gmail.com; 8Department of Pathology and Molecular Immunology, School of Medicine & Biomedical Sciences, University of Porto (ICBAS-UP), 4050-313 Porto, Portugal; 9Internal Medicine Department, Póvoa de Varzim/Vila do Conde Hospital Centre, 4490-421 Póvoa de Varzim, Portugal; 10FP-I3ID—Instituto de Investigação, Inovação e Desenvolvimento, FP-BHS—Biomedical and Health Sciences Resarci Unit, FFP—Fundação Fernando Pessoa, 4249-004 Porto, Portugal

**Keywords:** breast cancer, molecular subtype, obesity, cancer cell metabolism, Warburg effect, glycolysis, gluconeogenesis

## Abstract

**Simple Summary:**

Breast cancer is a complex pathology characterized by several features including molecular subtype (MS). Immunohistochemistry assays were used to investigate the expression of enzymes involved in glycolysis and gluconeogenesis. The analysis involved stratifying the data based on MS, body mass index (BMI), and the combination of BMI with MS (mBMI). This study revealed significant differences in the expression of three specific enzymes—pyruvate carboxylase (PC), phosphoenolpyruvate carboxykinase (PCK), and fructose-1,6-bisphosphatase (FBP)—among tumor cells when stratified by MS and mBMI. Moreover, the expression levels of these enzymes were found to be closely related to hormonal receptor and HER2 status, as well as correlated pathological stage and histological grade. Obesity appeared to have an impact on these differences, particularly in the expression of PC. However, it was observed that these differences were not influenced by the presence of adipocyte deposition or inflammatory infiltration within the tumor microenvironment. Nevertheless, the expression of PCK and FBP was also influenced by the presence of obesity-related conditions such as diabetes, hypertension, and dyslipidemia. In summary, this study highlights the existence of distinct metabolic profiles for breast cancer based on its molecular subtypes, and how these profiles are affected by obesity and related health conditions.

**Abstract:**

Breast cancer is a heterogeneous entity, where different molecular subtypes (MS) exhibit distinct prognostic and therapeutic responses. A series of 62 breast cancer samples stratified by MS was obtained from the tumor biobank of IPO-Porto. The expression of glycolysis and gluconeogenesis-regulating enzymes was investigated by immunohistochemistry. Data analysis included stratification according to MS, body mass index (BMI), and BMI with MS (mBMI). We observed significant differences in pyruvate carboxylase (PC), phosphoenolpyruvate carboxykinase (PCK), and fructose-1,6-bisphosphatase (FBP) tumor cell expression when stratified by MS and mBMI. The expression of these enzymes was also statistically dependent on hormonal receptors and HER2 status and correlated with pathological stage and histological grade. Obesity tended to attenuate these differences, particularly in PC expression, although these were not affected by adipocyte deposition or inflammatory infiltration at the tumor microenvironment. Nonetheless, PCK and FBP expression was also modified by the presence of obesity-associated disorders like diabetes, hypertension, and dyslipidemia. Taken together, these findings identify metabolic fingerprints for breast cancer as distinct histological types, which are affected by the presence of obesity and obesity-associated conditions. Despite the biological role of the differential expression of enzymes remaining unknown, the current study highlights the need to identify the expression of gluconeogenic-regulating enzymes as a tool for personalized medicine.

## 1. Introduction

Cancer and obesity are two noncommunicable diseases considered a global epidemic with a high impact on health outcomes [1]. Obesity is associated with several types of cancer including breast cancer (BC) [2], a complex and heterogeneous disease, regarding tumor classification, prognostic factors, therapeutical strategies, and clinical outcomes [3]. Since both diseases present increasing incidence rates [4,5], it is of paramount importance to understand how obesity affects breast cancer.

The obesity–breast cancer link has been extensively studied. Several mechanisms have been proposed to contribute to this connection. Among them, the local and systemic effect of the low chronic state of inflammation characteristic of obese patients; the dysregulated concentrations of adipokines like leptin and adiponectin; the disrupted insulin/IGF signaling pathway; and the most straightforward association, the expression of aromatase in the adipose tissue, which is able to convert androgens into estrogen, are the most established ones [6]. High estrogen levels were found to contribute to tumorigenesis and tumor progression by its capacity to induce DNA damage and promote angiogenesis and cell proliferation in hormone-dependent breast cancers [7]. Adipocytokines, responsible for metabolic homeostasis, are also found dysregulated in obesity. and this dysregulation has been associated with metabolic syndrome conditions like diabetes, hypertension, and dyslipidemia [8].

Tumor cells may respond to the obese stimulation factors through the expression of particular receptors, including hormone receptors. Molecular subtype (MS) identifies breast cancer according to hormone receptors (estrogen and progesterone) and human epidermal growth factor receptor 2 (HER2) expression. Four MS: luminal A, luminal B, HER2-enriched, and triple negative are widely accepted. This classification is of extreme importance both for BC prognosis as well as therapeutic targets, namely endocrine therapy for luminal A and B or anti-HER2 monoclonal antibodies against HER2+ breast cancer [9].

*Reprogramming energy metabolism* has been considered a hallmark of cancer since 2011 [10]. It reflects the tumor cell ability to modify or reprogram the metabolism to most efficiently support its proliferation [10]. As first described by Otto Warburg, tumor cells reprogram their glucose metabolism to an aerobic glycolysis, hence surpassing oxidative phosphorylation even in the presence of oxygen, the well-known *Warburg effect* [11]. Several variations and observations of this mechanism have been uncovered ever since. Previous studies identified two metabolic different populations, working symbiotically in the tumor microenvironment: one population secreting lactate, and the other consuming lactate [12]. Another recent proposition describes that cancer-associated fibroblasts consume lactate and pyruvate, producing glucose. In turn, this glucose is captured by tumor cells, thus promoting an efficient production of essential energy to meet the proliferative needs of both cells. This self-sustaining “Cori cycle” between the microenvironment stroma and the tumor cell is called the *reverse Warburg effect* [13]. In 2019, our group unraveled another metabolic reprograming in breast cancer cell lines. Under obesity-mimicking conditions, we observed a switch in the central metabolic pathway, which we termed *Warburg effect inversion*. MCF-7 breast cancer cells in the presence of adipocyte-secreted medium consumed lactate-originating glucose as observed by metabolomic assays. This inversion of tumor cell metabolism was concomitant with increased tumor cell aggressiveness in vitro [14].

The present work aims to explore glycolysis and gluconeogenesis rate-limiting enzymes in a human breast cancer series. The expression of glycolysis and gluconeogenesis rate-limiting enzymes, catalyzing irreversible reactions, was quantified in tumor samples from different molecular subtypes. The results were correlated with body mass index (BMI) and obese-associated characteristics such as clinical parameters, adipose tissue infiltration, nonspecific inflammation status, and metabolic syndrome conditions.

## 2. Materials and Methods

### 2.1. Study Population and Ethical Approval

The study was performed according to the principles of the Declaration of Helsinki. Ethical approval was obtained by the institutional ethics committee of the Portuguese Oncology Institute of Porto (IPO-Porto) Francisco Gentil, Porto, Portugal. All participants gave their informed consent.

Tumor samples of female patients diagnosed in IPO-Porto were obtained from the IPO biobank. All patients were submitted to conservative surgery or mastectomy between 2017 and 2019. Inclusion criteria included no neoadjuvant treatment, tumors with no special type (NST—Invasive ductal carcinoma), the pathological stage between I-III, and with PROSIGNA/PAM50 genetic study performed for molecular subtypes.

### 2.2. Sample Characteristics

Body mass index (BMI) was calculated according to the Quetelet index [15] and samples ranked in normal weight (18.5 ≤ BMI < 25 kg/m^2^) and obese (BMI ≥ 30 kg/m^2^), as calculated before surgical intervention. PROSIGNA/PAM50 genetic study was performed to assess the MS. Tumors were classified according to their hormonal and HER2 status as positive or negative. Cancer staging (pathologic stage) was assessed accordantly to the *American Joint Committee* on Xancer by the tumor node metastasis (TNM) system [16]. Histological differentiation grade was classified according to the *Nottingham histological grade*, a combination of nuclear grade, mitotic rate, and tubule formation [17]. Topographic localization was divided into quadrants: the inferior outer quadrant (IOQ), inferior inner quadrant (IIQ), superior outer quadrant (SOQ), superior inner quadrant (SIQ), central quadrant (CQ), multiple (present in more than one quadrant), or other (axillary and nipple). Diabetes, hypertension, and dyslipidemia were assessed at the time of the first medical appointment in IPO-Porto. Ki67 proliferation biomarker was quantified by immunohistochemistry according to internal protocols, stratified in three groups (<15%; 15–30%; >30%).

### 2.3. Hematoxylin–Eosin Staining

Hematoxylin and eosin (H&E) (Hematoxylin: Cat. 72704; Thermo Fisher Scientific, Waltham, MA, USA; Eosin: Cat. 71211; Thermo Fisher Scientific, Waltham, MA, USA) staining was performed. Adipose tissue infiltration and nonspecific inflammation were quantified anonymously and independently by three researchers. Each trait was scored as follows: 0. absent; 1. scarce; 2. moderate; and 3. abundant.

### 2.4. Immunohistochemistry Assays

Paraffin-embedded sections were deparaffinized and hydrated. Sections were submitted to heat antigenic recovery in citrate buffer. Endogenous peroxidase activity and nonspecific binding were blocked. Tissue sections were allowed to react with primary/secondary pairing antibodies. Immunoreactivity was visualized with avidin–biotin reaction (Avidin/Biotin Blocking Kit #SP-2001, Vector Laboratories, Newark, NJ, USA) and revelation was performed using DAB substrate (Pierce™ DAB Substrate Kit #34002, Thermo Fisher Scientific, Waltham, MA, USA) counterstained with haematoxylin (#72704; Thermo Fisher Scientific, Waltham, MA, USA).

The isoforms under analysis were hexokinase (HK)—HK II; phosphofructokinase (PFK)—PFK L; pyruvate Kinase (PK)—PK M; pyruvate carboxylase (PC)—PC B; phosphoenolpyruvate carboxykinase (PCK)—PCK 2; fructose-1,6-bisphosphatase (FBP)—FBP 1; glucose-6-Phosphatase (G6P)—G6P alpha. Specifications of each assay are described in the Appendix A and results from the positive control can be visualized in the Appendix A. A negative control was performed with incubation with a blocking solution (without primary antibody).

All stained specimens were visualized under a Nikon Eclipse 501 optical microscope (Nikon, Tokio, Japan). Quantification regarding enzymatic expression in tumor cells was anonymous and independently scored by three researchers. The area was classified according to the percentage of positive cells, as follows: 0. unstained; 1. less than 5%; 2. from 5 to 50%; and 3. over 50% stained. Intensity was scored as: 0. negative; 1. weak; 2. moderate and 3. strong. The score was obtained by multiplying the area and intensity on a scale of 0 to 9. The final score was categorized as follows: No expression = score of 0; low expression = score of 1, 2, and 3, and high for scores of 4, 6, and 9.

### 2.5. Statistical Analysis

The analysis included stratification according to BMI (Normal weight vs. Obese patients), MS (luminal A vs. luminal B vs. triple negative vs. HER2+), and BMI with MS, further presented as mBMI (luminal A normal weight vs. luminal A obese vs. luminal B normal weight vs. luminal B obese vs. triple-negative normal weight vs. triple-negative obese vs. HER2+ normal weight vs. HER2+ obese).

Statistical analysis was performed using IBM^®^ SPSS^®^ version 27 software (IBM, Endicott, NY, USA). Continuous variables were tested for normality by the Kolmogorov–Smirnov test. Descriptive statistics was used for data description in terms of absolute frequencies and valid percentages. Continuous variables were presented as mean values and standard deviation. The chi-square (χ^2^) test was used to analyze enzymatic expression, adipose tissue infiltrations, and nonspecific inflammation with BMI, MS, and mBMI. The correlation between tumor characteristics/obese-related features and enzymatic expression was also examined using Spearman’s correlation coefficient (r) with two tails. The *p*-value inferior to 0.05 was considered statistically significant.

## 3. Results

### 3.1. Demographic and Clinicopathological Data

A total of 62 samples were included in this study, 32 from normal-weight patients (8 luminal A; 8 luminal B; 8 triple negative; 8 HER2+) and 30 from obese patients (8 luminal A; 8 luminal B; 8 triple negative; 6 HER2+). Table 1 displays the demographic and clinicopathological information from the patients and the *p*-value from chi-square analysis of MS, BMI, and mBMI stratification.

As expected, a significant association was found between the pathological stage (*p*-value = 0.037), histological grade (*p*-value: <0.001), and MS, which was also evident with mBMI. Luminal cases are frequently associated with lower pathological stages and histological grades. Luminal A specifically expresses lower levels of proliferation genes which translates to a good prognosis [18]. MS stratification further revealed a significantly higher percentage of carcinoma in situ presenting necrosis (*p*-value: 0.042), which is lost in mBMI stratification. Apart from histological grade, none of the included features was significantly different in obese versus nonobese patients within each MS. A more in-depth insight from the results of MS, BMI, and mBMI stratification can be consulted in Appendix A.

### 3.2. Expression of Glycolysis and Gluconeogenesis Rate-Limiting Enzymes

The tumor cell expression of enzymes catalyzing the irreversible reactions of glycolysis (HK, PFK, and PK) and gluconeogenesis (PC, PCK, FBP, and G6P) was investigated by immunohistochemistry. Examples of absence, low expression, and high expression patterns are illustrated in Appendix A.

It is interesting to note that no significant difference was observed in any of the glycolytic enzymes studied regarding MS stratification. HK expression was rather reduced for every MS, being absent in luminal B cases. Inversely, there were significant differences in most gluconeogenic enzymes, namely in PC, PCK, and FBP. Triple-negative subtypes did not express FBP in contrast to luminal A and B, where a significant number of cases overexpressed FBP but not PC and PCK. HER2+ cases displayed higher expression of PC and PCK.

When stratified by BMI, we observed a slight decreased expression in the three glycolytic enzymes in breast cancer tissue of obese individuals (although without statistical significance). Regarding gluconeogenic enzyme expression, there was a switch in the expression pattern in FBP and G6P. An increased number of obese patients overexpressed both these enzymes in tumors (again without statistical differences) in contrast to PC and PCK. A tendency towards downregulation of PC expression was observed in obese cases (*p* = 0.065) relative to normal-weight subjects.

We next investigated whether BMI influenced the metabolic characteristics of MS. Statistical differences observed in MS analysis were maintained although with lower statistical power, probably due to the smaller size of the sample. In most cases, PFK and PK glycolytic enzyme expression tended to decrease in obese patients of the different MS, except in triple-negative cases. Further, in comparison to normoponderal cases with the same MS, obese patients presented PC downregulation, and an upregulation of FBP (again, in this latter, with the exception of triple-negative cases) (Figure 1).

In summary, glycolytic enzyme expression did not differ either between MS or between BMI groups. PFK and PK expression was more abundant than HK. Expression of gluconeogenic enzymes (PC, PCK, FBP, and G6P) varied significantly with MS. A tendency to decrease PC was observed in breast cancer from obese patients.

### 3.3. Correlation between Rate-Limiting Enzymes Expression and Clinicopathological Features

We further assessed the correlation between the enzymatic score (0–9) and cancer-associated clinicopathological features. The results are shown in Table 2. HK was positively correlated with the HER2 receptor status and with the presence of vascular invasion. PK was solely correlated with the pathological stage. PC was negatively correlated with BMI, ER, and PR and positively correlated with the pathological stage, histological grade (including histological grade mitosis assessment), and HER2. PCK was negatively correlated with ER and PR and positively correlated with HER2. An additional positive correlation was found with histological grade mitosis assessment. Inversely, FBP was positively correlated with hormonal receptors and negatively correlated with histological grade (including nuclear grade and mitosis of histological grade assessment).

These findings indicate that rate-limiting enzymes of the gluconeogenic pathway are potential prognostic factors in breast cancer, FBP a marker of good prognosis, and PC and PCK markers of bad prognosis.

### 3.4. Intratumor Adipose Tissue Deposition

Adipocytes influence the central metabolism by releasing adipokines, cytokines, fatty acids, and growth factors, among others [19]. We next investigated whether the presence of tumor microenvironment (TME) adipocytes accompanied the observed differences in enzymatic expression in tumor cells. No statistical significance was achieved in the number of adipose tissue depots when stratified either by BMI, MS, or mBMI (Figure 2).

Adipose tissue infiltration was not associated with tumor cells’ enzymatic expression (Table 3). Interestingly, we found PC expression to be the only enzyme with a positive correlation with adipose tissue infiltration within TME.

### 3.5. Inflammatory Infiltration

Obesity is associated with increased inflammation. Thus, we next examined the intratumoral presence of inflammatory infiltration in our series of breast cancer tissue (Figure 3). No difference was observed regarding the extent of inflammatory infiltration in breast cancer series upon BMI, MS, or mBMI stratification. Triple-negative and HER2+ cases exhibit more abundant inflammatory infiltration than luminal A and B, still without significant differences. Comparison between BMI and normal weight for each MS (Figure 3d) revealed that a more abundant inflammatory infiltration was observed in luminal B and Her2+ normal weight cases, and in obese triple-negative cases.

Statistical results between inflammation and enzymatic expression are displayed in Table 4. Tumor inflammatory infiltration was statistically associated with PC upregulation in tumor cells. Spearman correlation revealed a tendency to a positive correlation between inflammation and PC expression.

### 3.6. Correlation between Tumor Cell Enzymatic Expression and Obesity-Associated Conditions

Diabetes, hypertension, and dyslipidemia cluster together with obesity, under the umbrella of metabolic syndrome. Table 5 illustrates the presence of diabetes, hypertension, and dyslipidemia in breast cancer patient series. The only statistically significant association was found between BMI and hypertension. Of the 25 individuals with hypertension, 17 were obese. Detailed data from this analysis can be consulted in Appendix A.

We further assessed the correlation between enzyme expression and obesity-related conditions. A positive correlation was observed between PCK and diabetes and between FBP and both hypertension and dyslipidemia (Table 6).

## 4. Discussion

To our knowledge, this is the first study addressing the expression of glycolytic and gluconeogenic rate-limiting enzymes in the molecular subtypes of breast cancer, and the potential influence of obesity on the expression of these enzymes.

As expected, a significant association was found between pathological stage, histological grade, and MS. In fact, luminal cases are associated with lower pathological stages and histological grades [18]. Moreover, with the exception of histological grade, none of these parameters was significantly different in obese versus nonobese patients within each MS (mBMI), being the difference in the histological grade being attributed to the molecular subtype itself. Additionally, the presence of necrotic tissue within in situ carcinoma was also statistically associated with molecular subtypes, in agreement with previous results [20].

We observed a reduction in glycolytic enzyme expression in obese cases, although without statistical significance. Recent evidence suggests that glucose in highly proliferating cells undergoes glycolysis to provide building blocks for anabolic pathways [21], such as the pentose phosphate pathway (PPP) that produces NADPH and ribose, hexosamines, glycerol-3-phosphate, and nonessential amino acids (Figure 4). Therefore, the decrease in glycolysis observed in obese patients with luminal and HER2+ breast cancers may be compensated by an increase in specific enzymes of the gluconeogenic pathway. That can be the case of FBP, an enzyme converting fructose-1,6-bisphosphate into fructose-6-phosphate, which then isomerizes to glucose-6-phosphate, the substrate of the PPP. Remarkably, obese triple-negative breast cancer patients behave differently. These tumors depend primordially on glycolysis for metabolism, and do not express FBP at all, either in obese or in nonobese patients. Similar findings have been reported in prior studies, with TNBC exhibiting either low FBP expression or the complete absence of expression [22]. It has been suggested that p53 deleterious mutations may play a role in suppressing FBP1 expression within this molecular subtype [23]. However, it is worth noting that there are studies presenting conflicting results, indicating that FBP1 expression levels in TNBC can be comparable to those observed in HER2+ [24]. We also found that FBP correlates positively with hormone receptors and negatively with histological grades.

It is interesting to note that PC, another gluconeogenic enzyme, does not follow the same trend; rather, it is significantly decreased in obese cases in every luminal, TN, and HER2+ breast cancer. The expression of PC in breast cancer can vary based on the molecular subtype. For instance, the HER2+ highly invasive cell line (MDA-MB-435) exhibits the highest levels of PC expression, which aligns with the findings of our study [25]. PC catalyzes the conversion of pyruvate into oxalacetate, the initial step of gluconeogenesis. The downregulation of this mitochondrial enzyme may be due to impaired mitochondrial or pyruvate metabolism, common features in tumor cells [26]. This enzyme is probably underestimated, given its correlation with other poor prognostic clinicopathological features. PC is negatively correlated with hormone receptors and positively with HER2. It is also associated with mitosis. Moreover, PC is the only enzyme that is significantly associated with tumor inflammatory infiltration. PCK expression is significantly increased in HER2+ cases from our cohort, contradicting previous results. In a study involving 177 breast cancer patients, stratified based on receptor status, a significant association was observed solely with the estrogen receptor and not with HER2 [27]. It was previously described that PCK leads to increased glucose production, promoting hyperglycemia and worsening the diabetes condition [28]. These findings corroborate the positive correlation between diabetes and PCK described in Table 6. PCK achieves statistical significance when stratified both by MS and mBMI, being also statistically positively correlated with HER2 and negatively with hormonal receptors and mitosis (similar to PC).

Our findings suggest that overexpression of PC and/or PCK is associated with poor prognosis. On the other hand, FBP is a marker of good prognosis in breast cancer. Altogether, the current study further provides evidence that gluconeogenic enzymes present prognostic value.

Obese and nonobese patients present similar rates of inflammatory infiltration and adipocytes within the tumor. These results were rather striking, since it is well established that obese people often display systemic low-grade chronic inflammation [29]. Nevertheless, this systemic inflammatory environment does not seem to play a crucial role in the local inflammation of the neoplastic tissue in the current analysis. In fact, comparing the different MS, the percentage of tumors lacking adipocytes was always lower in obese patients than in normal-weight ones, except in the case of triple-negative tumors. This decrease in the number of adipocytes within tumor tissue in obese subjects may just be a consequence of an excess of fat in normal adjacent breast tissue that plays a paracrine role. Previous studies observed that adipocytes can undergo phenotypic alterations when exposed to tumor cells, termed cancer-associated adipocytes (CAA) [30]. CAA is mostly associated with increased catabolism, with the release of metabolites such as pyruvate, lactate, free fatty acids, and ketone bodies. We recognized that our adipose tissue deposition results could be, to some extent, limited, given the adipocyte dedifferentiation to a fibroblast-like phenotype, which could be confused with other stromal cells. Nonetheless, the importance of differentiated adipocytes in the TME should not be underestimated. However, several questions remain to be clarified. Given the well-established metabolic, inflammatory, and endocrine role of adipose tissue [31], it is important to know whether this switch in cancer cell metabolism is caused by inflammatory cytokines, adipokines (leptin, adiponectin, resistin) or a mere enhanced energy availability, as adipocyte-derived fatty acids may be used as energy source in tumor cells.

We addressed the expression of glycolysis and gluconeogenesis-regulating enzymes in breast cancer-stratified cases by MS and by BMI. Nonetheless, there are certainly several other players in this crosstalk. Adipose tissue-releasing hormones, adipocytokines, and oxidative stress are also of paramount importance [19,32]. Moreover, menopausal status is also a crucial player when the sample population is stratified by BMI. It is previously described that obesity may be associated with a reduced risk of breast cancer before menopause and with increased risk in postmenopausal women, associated with hormonal receptors and MS [33]. Although relevant, such features are beyond the scope of the current study. Additionally, we cannot rule out the effect of oncogenes and tumor suppressor genes that control metabolic cues within the tumor cell. For instance, TIGAR is a glycolysis regulator that is controlled by p53 [34]. A deleterious mutation of this tumor suppressor gene interferes with metabolism, ultimately resulting in cell behavior changes.

We are aware of the study’s limitations. We believe that an increased number of samples could improve the statistical strength of the study and uncover additional differences. Moreover, immunohistochemistry is a limited technique, prone to human errors and lacking standardized protocols for research. Nonetheless, we believe that this study brought new insights regarding the expression of enzymes of two main metabolic pathways in human breast cancer, paving the way towards a precise medicine. Further studies are needed to deeper highlight the precise role of these enzymes in breast tumor cells, and the potential therapeutic targets.

## 5. Conclusions

This study reveals that each molecular subtype of breast cancer holds a particular gluconeogenic fingerprint, which ultimately adapts metabolism to the needs of the tumor cell. PC and PCK are overexpressed in HER2+ cases, in contrast to the other MS. On the other hand, FBP was overexpressed in luminal groups, and absent in triple-negative ones.

In obese patients, however, an increase in FBP expression has been observed in luminal A, B, and HER2+ MS. PC expression was decreased upon obesity, but this trend was not found in PCK. These results are not associated with variations in TME inflammation or adipocyte depots, since no changes were observed in obese vs. nonobese patients for any MS.

In addition, gluconeogenic enzymes such as FBP, PC, and PCK correlated differently with pathological stage, histological grade, HER2, and hormonal receptors, suggesting a potential prognostic value of these enzymes. Additionally, PCK is significantly correlated with diabetes and FBP is significantly correlated with hypertension and dyslipidemia, implying that not only obesity, but rather other obesity-associated disorders must be taken into account in breast cancer prognosis and treatment.

## Figures and Tables

**Figure 1 cancers-15-04936-f001:**
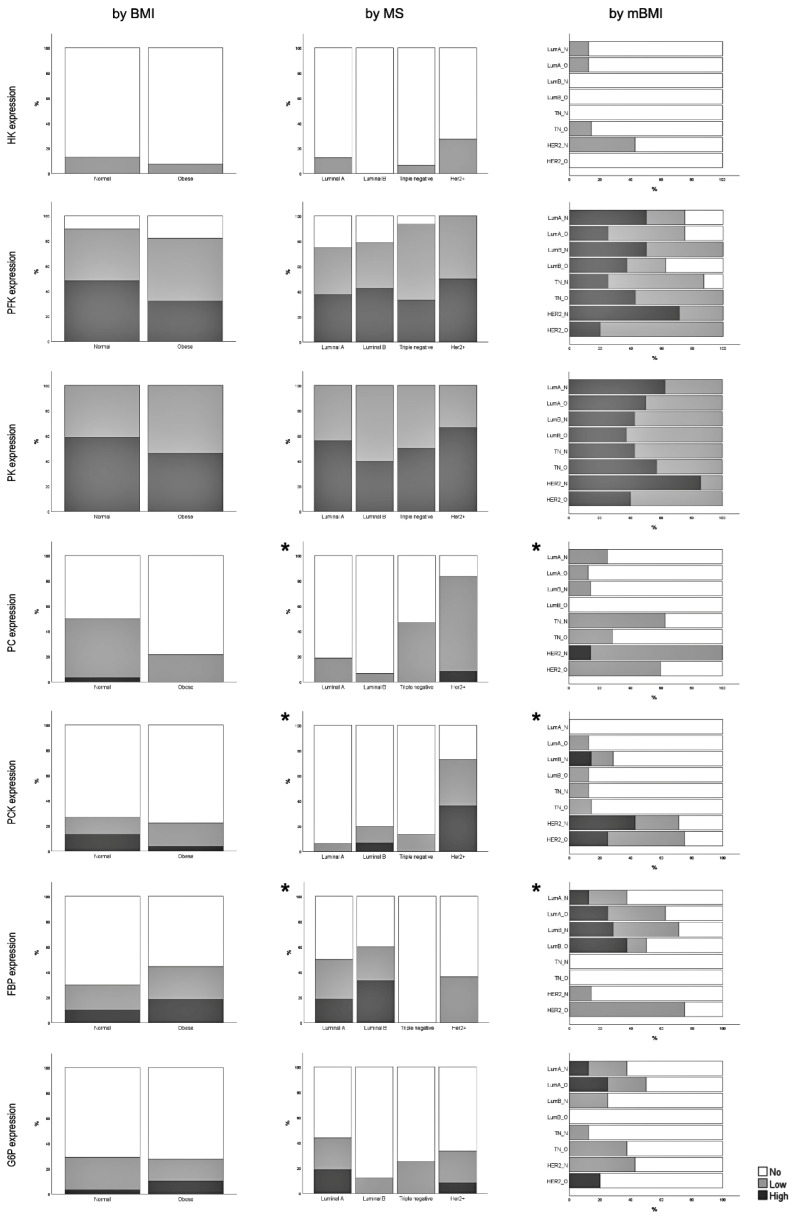
Glycolytic and gluconeogenic rate-limiting enzyme expression assessed by immunohistochemistry stratified by BMI, MS, and mBMI. No significant differences between normal weight and obese patients were observed. [HK χ^2^ test *p*-value: 0.467; PFK χ^2^ test *p*-value: 0.422; PK χ^2^ test *p*-value: 0.357; PC χ^2^ test *p*-value: 0.065; PCK χ^2^ test *p*-value: 0.410; FBP χ^2^ test *p*-value: 0.491; G6P χ^2^ test *p*-value: 0.438]; significant differences between molecular subtypes were observed in PC, PCK, and FBP. [HK χ^2^ test *p*-value: 0.148; PFK χ^2^ test *p*-value: 0.428; PK χ^2^ test *p*-value: 0.566; PC χ^2^ test *p*-value: 0.001 *; PCK χ^2^ test *p*-value: 0.002 *; FBP χ^2^ test *p*-value: 0.007 *; G6P χ^2^ test *p*-value: 0.262]; significant differences between molecular subtypes were observed in PC, PCK, and FBP; [HK χ^2^ test *p*-value: 0.134; PFK χ^2^ test *p*-value: 0.345; PK χ^2^ test *p*-value: 0.653; PC χ^2^ test *p*-value: 0.010 *; PCK χ^2^ test *p*-value: 0.047 *; FBP χ^2^ test *p*-value: 0.023 *; G6P χ^2^ test *p*-value: 0.304]; legend: HK—hexokinase; PFK—phosphofructokinase; PK—pyruvate kinase; PC—pyruvate carboxylase; PCK—phosphoenolpyruvate carboxykinase; FBP—fructose-1,6-bisphosphatase; G6P—glucose-6 phosphatase.

**Figure 2 cancers-15-04936-f002:**
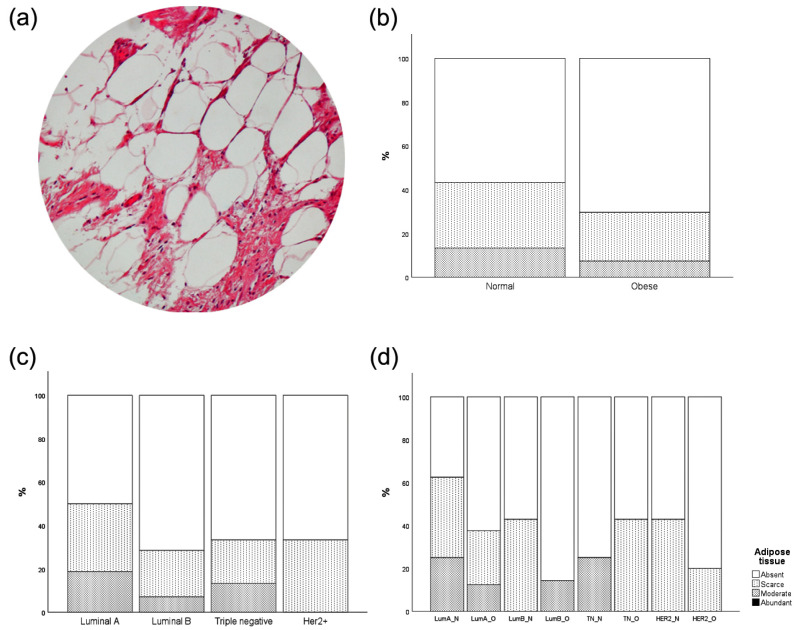
Quantification of intratumor adipose tissue deposition according to BMI, MS, or both (mBMI). No statistical differences were found. (**a**) Representative image of adipose tissue infiltration (amplification: 100X); (**b**) stratification by BMI, χ^2^ test *p*-value: 0.542; (**c**) stratification by MS χ^2^ test *p*-value: 0.687; (**d**) stratification by mBMI χ^2^ test *p*-value: 0.409; legend: LumA_N—luminal A normal weight; LumA_O—luminal A obese; LumB_N—luminal B normal weight; LumB_O—luminal B obese; TN_N—triple-negative normal weight; TN_O—triple-negative obese; HER2_N—HER2+ normal weight; HER2_O—HER2+ obese.

**Figure 3 cancers-15-04936-f003:**
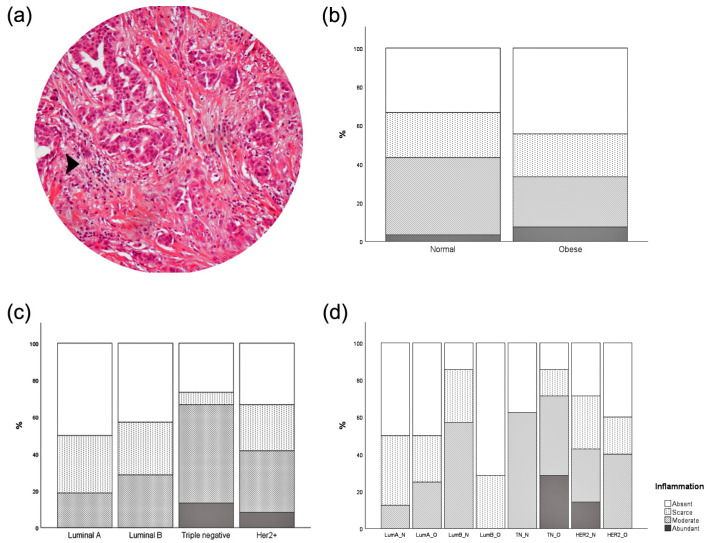
Quantification of inflammatory infiltration according to BMI, MS, or both (mBMI). No statistical differences were found. (**a**) Representative image of inflammatory infiltration (amplification: 100X). Black arrow indicates an example of inflammatory infiltration; (**b**) stratification by BMI, χ^2^ test *p*-value: 0.625; (**c**) stratification by MS χ^2^ test *p*-value: 0.330; (**d**) stratification by mBMI χ^2^ test *p*-value: 0.221; legend: LumA_N—luminal A normal weight; LumA_O—luminal A obese; LumB_N—luminal B normal weight; LumB_O—luminal B obese; TN_N—triple-negative normal weight; TN_O—triple-negative obese; HER2_N—HER2+ normal weight; HER2_O—HER2+ obese.

**Figure 4 cancers-15-04936-f004:**
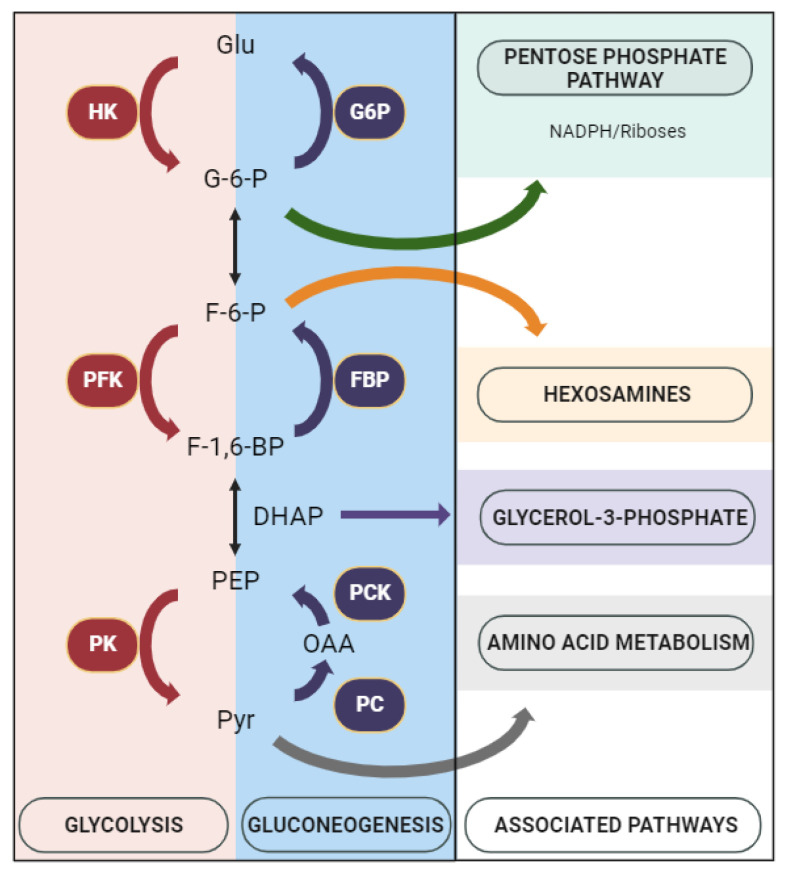
Glycolysis, gluconeogenesis, and associated pathways. Legend: Glu—glucose; G-6-P—glucose-6 phosphate; F-6-P—fructose 6 phosphate; F-1,6-BP—fructose 1,6 bisphosphate; DHAP—dihydroxyacetone phosphate; PEP—phosphoenolpyruvate; Pyr—pyruvate; OAA—oxaloacetate; HK—hexokinase; PFK—phosphofructokinase; PK—pyruvate kinase; PC—pyruvate carboxylase; PCK—phosphoenolpyruvate carboxykinase; FBP—fructose-1,6-bisphosphatase; G6P—glucose-6-phosphatase.

**Table 1 cancers-15-04936-t001:** Demographic and cancer-associated clinicopathological data from the breast cancer patient series stratified by BMI, MS, or both (mBMI).

*n* (%)	Total	*p*-Value BMI Analysis	*p*-Value MS Analysis	*p*-Value mBMI Analysis
	62 (100.0%)			
**Age at diagnosis**				
(Mean ± SD)	57.48 ± 10.53	0.480	0.218	0.604
**BMI categories**				
Normal weight	32 (51.6%)	-	0.974	-
Obese	30 (48.4%)			
**Molecular subtypes**				
Luminal A	16 (25.8%)	0.974	-	-
Luminal B	16 (25.8%)			
Triple negative	16 (25.8%)			
HER2+	14 (22.6%)			
**Topographic localization**				
CQ	3 (4.8%)	0.680	0.052	0.075
IOQ	4 (6.5%)			
SOQ	21 (33.9%)			
SIQ	4 (6.5%)			
Multiple	27 (43.5%)			
Other	3 (4.8%)			
**Laterality**				
Right	32 (51.6%)	0.806	0.632	0.075
Left	30 (48.4%)			
**Pathological stage**				
Stage I	36 (59.0%)	0.321	**0.037 ***	0.051
Stage II	22 (36.1%)			
Stage III	3 (4.9%)			
**Histological grade**				
Grade I Grade II	0 (0%) 37 (59.7%)	0.227	**<0.001 ***	**<0.001 ***
Grade III	25 (40.3%)			
**Carcinoma in situ**				
Presence	24 (38.7%)	0.469	0.282	0.357
Size. cm (mean ± SD)	1.35 ± 2.47	0.649	0.798	0.463
Extensive	20 (32.3%)	0.472	0.367	0.253
Microcalcifications	28 (45.2%)	0.818	0.236	0.333
Necrosis	39 (62.9%)	0.946	**0.042 ***	0.126
**Invasive carcinoma**				
Size. cm (mean ± SD)	1.76 ± 0.75	0.465	0.197	0.210
Multifocal	14 (22.6%)	0.891	0.564	0.656
Invasion	19 (30.6%)	0.511	0.661	0.783
**Metastatic nodes**				
0	41 (66.2%)	0.364	0.144	0.082
1–3	18 (29.0%)			
4–9	1 (1.6%)			
>10	2 (3.2%)			
Size cm (mean ± SD) (Larger metastasis)	11.48 ± 31.49	0.764	0.056	0.340

Legend: SD—standard deviation; CQ—central quadrant; IOQ—inferior outer quadrant; SOQ—superior outer quadrant; SIQ—superior inner quadrant; * statistical significance highlighted in bold (*p*-value < 0.05).

**Table 2 cancers-15-04936-t002:** Spearman correlation between enzymatic score and demographic and clinicopathological features.

	HK		PFK		PK		PC		PCK		FBP		G6P	
	r	*p*	r	*p*	r	*p*	r	*p*	r	*p*	r	*p*	r	*p*
**BMI**	−0.147	0.275	−0.162	0.229	−0.059	0.650	**−0.303**	**0.021 ***	−0.138	0.305	0.138	0.305	0.047	0.720
**ER**	−0.151	0.268	−0.177	0.192	−0.168	0.196	**−0.514**	**<0.001 ***	**−0.284**	**0.034 ***	**0.463**	**<0.001 ***	0.020	0.881
**PR**	−0.151	0.268	−0.177	0.192	−0.168	0.196	**−0.514**	**<0.001 ***	**−0.284**	**0.034 ***	**0.463**	**<0.001 ***	0.020	0.881
**HER2**	**0.277**	**0.039 ***	0.259	0.054	0.139	0.285	**0.503**	**<0.001 ***	**0.569**	**<0.001 ***	−0.075	0.580	0.059	0.658
**Ki67**	−0.045	0.799	0.122	0.486	0.123	0.463	0.089	0.604	0.288	0.093	−0.038	0.829	0.009	0.956
**TL**	0.043	0.752	0.185	0.169	0.132	0.306	0.105	0.433	−0.141	0.296	−0.209	0.119	−0.209	0.110
**LAT**	−0.086	0.524	0.113	0.401	0.131	0.312	−0.084	0.529	−0.028	0.834	0.156	0.246	−0.046	0.726
**pST**	0.053	0.698	0.138	0.310	**0.263**	**0.041 ***	**0.437**	**<0.001 ***	0.185	0.172	−0.215	0.111	0.154	0.246
**HG**	0.078	0.566	0.106	0.434	0.120	0.353	**0.452**	**<0.001 ***	0.238	0.075	**−0.396**	**0.002 ***	0.014	0.918
**HGnc**	0.117	0.384	0.198	0.139	0.137	0.287	0.249	0.060	0.193	0.149	**−0.304**	**0.021 ***	0.059	0.656
**HGm**	0.050	0.711	0.032	0.812	0.034	0.796	**0.392**	**0.002 ***	**0.272**	**0.041 ***	**−0.320**	**0.015 ***	−0.007	0.955
**HGtf**	−0.083	0.541	−0.021	0.875	0.090	0.489	0.243	0.066	0.036	0.793	−0.144	0.286	−0.020	0.881
**ISs**	−0.016	0.906	−0.120	0.373	−0.094	0.469	−0.071	0.595	−0.151	0.263	0.088	0.514	0.230	0.077
**ISext**	0.026	0.846	−0.027	0.840	−0.063	0.624	−0.166	0.212	−0.173	0.198	0.142	0.293	0.212	0.194
**ISmc**	0.122	0.365	−0.082	0.546	−0.187	0.146	−0.148	0.267	−0.143	0.289	0.183	0.172	0.063	0.631
**ISnec**	0.154	0.254	0.098	0.468	−0.091	0.483	0.196	0.139	0.025	0.854	0.032	0.813	0.026	0.842
**INVs**	−0.016	0.907	0.043	0.753	0.155	0.229	0.242	0.067	0.055	0.683	−0.237	0.075	0.210	0.108
**INVm**	0.234	0.080	−0.137	0.309	−0.053	0.680	−0.031	0.816	0.014	0.918	−0.014	0.920	0.109	0.407
**INVvi**	**0.269**	**0.043 ***	0.160	0.234	0.094	0.467	0.077	0.567	−0.027	0.840	−0.012	0.930	0.059	0.656
**METs**	−0.090	0.504	0.089	0.509	−0.093	0.474	0.048	0.720	−0.154	0.253	0.009	0.947	0.076	0.564

Legend: r—spearman correlation factor; *p*—*p*-value; HK—hexokinase; PFK—phosphofructokinase; PK—pyruvate kinase; PC—pyruvate carboxylase; PCK—phosphoenolpyruvate carboxykinase; FBP—fructose-1,6-bisphosphatase; G6P—glucose-6-phosphatase; BMI—body mass index; ER—estrogen receptor; PR—progesterone receptor; HER2—human epidermal growth factor receptor 2; TL—topographic localization; LAT—laterality; pST—pathological stage; HG—histological grade; HGng—nuclear grade for histological classification; HGm—mitosis for histological classification; HG—tubule formation for histological classification; ISs—in situ size; ISext—in situ extensive; ISm—in situ microcalcifications; ISnec—in situ necrosis; INVs—invasive size; INVmi—invasive carcinoma multifocality; INVvi—vascular invasion; METs—metastatic size; * statistical significance highlighted in bold (*p*-value < 0.05).

**Table 3 cancers-15-04936-t003:** Chi-square analysis and spearman correlation between adipose tissue infiltration and enzymatic expression.

Enzymes	χ^2^ Test	Spearman Correlation	
	*p*-Value	R	*p*-Value
HK	0.635	−0.038	0.784
PFK	0.509	−0.054	0.690
PK	0.318	−0.113	0.403
PC	0.580	0.035	0.796
PCK	0.590	−0.111	0.415
FBP	0.222	−0.190	0.160
G6P	0.569	−0.049	0.718

Legend: r—spearman correlation factor; HK—hexokinase; PFK—phosphofructokinase; PK—pyruvate kinase; PC—pyruvate carboxylase; PCK—phosphoenolpyruvate carboxykinase; FBP—fructose-1,6-bisphosphatase; G6P—glucose-6-phosphatase.

**Table 4 cancers-15-04936-t004:** Chi-square analysis and Spearman correlation between inflammatory infiltration and enzymatic expression.

Enzymes	χ^2^ Test	Spearman Correlation	
	*p*-Value	r	*p*-Value
HK	0.411	0.209	0.126
PFK	0.739	0.117	0.391
PK	0.264	−0.017	0.900
PC	**<0.001 ***	0.248	0.062
PCK	0.411	0.179	0.186
FBP	0.465	−0.071	0.604
G6P	0.938	0.097	0.471

Legend: r—spearman correlation factor; HK—hexokinase; PFK—phosphofructokinase; PK—pyruvate kinase; PC—pyruvate carboxylase; PCK—phosphoenolpyruvate carboxykinase; FBP—fructose-1,6-bisphosphatase; G6P—glucose-6-phosphatase. * Statistical significance highlighted in bold (*p*-value < 0.05).

**Table 5 cancers-15-04936-t005:** Chi-square test of obesity-associated pathologies (diabetes, hypertension, and dyslipidemia) stratified according to BMI, MS, and mBMI.

*n* (%)	Total	*p*-Value BMI Analysis	*p*-Value MS Analysis	*p*-Value mBMI Analysis
	62 (100.0%)			
**Diabetes**				
Yes	7 (11.5%)	0.589	0.883	0.791
No	54 (88.5%)			
**Hypertension**				
Yes	25 (41.0%)	**0.008 ***	0.949	0.088
No	36 (59.0%)			
**Dyslipidemia**				
Yes	39 (63.9%)	0.175	0.495	0.549
No	22 (36.1%)			

Note: Data presented in valid percentage (missing data from 1 patient). * Statistical significance highlighted in bold (*p*-value < 0.05).

**Table 6 cancers-15-04936-t006:** Spearman correlation between enzymatic score and obesity related features (diabetes, hypertension, and dyslipidemia).

Enzymes	Diabetes		Hypertension		Dyslipidemia	
	r	*p*-Value	r	*p*-Value	r	*p*-Value
HK	−0.131	0.337	−0.065	0.635	0.079	0.564
PFK	0.070	0.607	0.177	0.187	0.180	0.180
PK	−0.099	0.447	0.025	0.849	0.029	0.822
PC	−0.035	0.794	−0.142	0.287	0.040	0.768
PCK	**0.304**	**0.022 ***	0.189	0.159	0.157	0.244
FBP	0.057	0.676	**0.320**	**0.015 ***	**0.328**	**0.013 ***
G6P	0.240	0.065	0.082	0.531	0.149	0.257

Legend: r—spearman correlation factor; HK—hexokinase; PFK—phosphofructokinase; PK—pyruvate kinase; PC—pyruvate carboxylase; PCK—phosphoenolpyruvate carboxykinase; FBP—fructose-1,6-bisphosphatase; G6P—glucose-6-phosphatase. * Statistical significance highlighted in bold (*p*-value < 0.05).

## Data Availability

The datasets generated during and/or analyzed during the current study are not publicly available due to privacy or ethical restrictions.

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
