# Peer review of "Breast Cancer Molecular Subtypes Differentially Express Gluconeogenic Rate-Limiting Enzymes—Obesity as a Crucial Player"

_cancers, 2023, doi:10.3390/cancers15204936_

Round 1

Reviewer 1 Report

This manuscript studies the relationship between expression of rate-limiting gluconeogenic enzymes and breast cancer molecular subtypes in the context of obesity. Overall, this paper would contribute new information for researchers interested in the intersection of breast cancer, obesity, and cancer metabolism.

The manuscript is overall well written. There are some formatting errors that are making it difficult to interpret tables. Table 2 for example is not formatted properly in the manuscript, so it isn’t clear which rows correspond to which clinicopathological features. This makes it impossible to confirm results match the conclusions.

I am unclear about the originality and novelty of study. The researchers are working with a new set of patient samples (albeit limited in size), which is valuable. However, there are publications which explore differences in cancer cell metabolism across breast cancer subtypes in a number of contexts. That are not properly referenced nor well integrated in the results and discussion. While these publications do not focus on the gluconeogenic enzymes, they include in their analysis, expression of the rate limiting enzymes such as F1,6BP (i.e., “Breast Cancer Subtypes Underlying EMT-Mediated Catabolic Metabolism, Metabolic Footprints and Molecular Subtypes in Breast Cancer”.

There is a conflicting result in comparing this manuscript to previous publications is the result described in 3.1 Expression of glycolysis and gluconeogenesis rate limiting enzymes, page 6, first para: “Triple negative did not express FBP in constrast to lumninal A and B, where a significant number of cases overexpressed FBP but not PC and PCK.” This previous publication https://www.hindawi.com/journals/dm/2017/7687851/ shows an increased in arbitrary units (relative amount) of F1,6BP (FBP) in the TNBC subtype compared to luminal. This paper isn’t even cited, and it seems that multiple papers related to this research area are not cited in this manuscript. 

While stratification by BMI is a common metric for investigating obesity, it presents an incomplete picture of a patient’s adipose levels. Additionally, previous publications have explored BMI as a risk factor in breast cancer, and have suggested differences in obesity association with specific molecular subtypes for pre- & post-menopausal women. It would be worth including if/how the expression of the gluconeogenic enzymes correlate differently across pre- & post-menopausal women.

The statistics seem to be appropriate. While the MS, BMI, and mBMI stratification groups meet the minimum number of samples required for chi-squared testing the Fishers test could be an alternative testing approach. 

Reviewer 2 Report

Manuscript ID: cancers-2607593

Based on the frequent co-morbidity between obesity and breast - Luis and coworkers examined the relation between obesity and glycolysis and gluconeogenesis rate-limiting enzymes in 62 human breast cancers. 

Changes on glucose uptake and metabolism is a hallmark of a variety of cancers and may be associated with reduced overall survival and a number of studies have indicated that obesity predispose to cancer but the mechanism is still uncertain. Finally, it is worth mentioning that FDG uptake is widely used as tracer in PET imaging in order to identify malignant cells. 

Metabolism was characterized by immunohistochemistry of 7 essential enzymes involved glucose metabolism - HK, PFK, PK, PC, FBP, G6P. Staining was correlated to a number of clinic pathological phenotypes including the BMI and the type of breast cancer and their staging. 

The study is mainly observational and at this stage the authors does not report any causal relations between obesity and cancer subtypes and survival. Nevertheless, the observations are important and some of the data are really intriguing - in particular the apparent lack FBP1 in basal cancers and the relatively high levels of G6PD in mAPO cancers (CIT subtype). 

The manuscript is well-written and easy to follow and the authors carefully balance the conclusions from the data. The experiments are technically sound and the clinical data are meticulous and relevant. This reviewer was impressed by the transparent and detailed supplemental tables that comprise a wealth of information. In the last paragraph of the discussion, it is mentioned in the last paragraph of the discussion, that immunohistochemistry may have limitations with respect to accuracy and quantification but we may also appreciate, that for once - in the days of genomics, we measure proteins. The comment however prompted me to test the expression of the enzymes on our patient breast cancer RNAseq data and the conclusions are clearly supported by these data.

As a minor comment I would ask if the authors could reconsider the nomenclature of the employed enzymes. I guess that the names refer to the antibody specificity but there are a number of isoforms for several of the enzymes and I wonder which you actually measure. As shown above in the genelist PFKL is is different from PFKM. 

In conclusion this is an interesting and important study that may pave the way for additional causal studies.
